# Characterisation of Macroinvertebrate Communities in Maritsa River (South Bulgaria)—Relation to Different Environmental Factors and Ecological Status Assessment

**Emilia Varadinova** [1,2,*]**, Lidia Sakelarieva** [1] **, Jiyoung Park** [1] **, Miroslav Ivanov** [1] **and Violeta Tyufekchieva** [2]

[1]    Department of Geography, Ecology and Environmental Protection, Faculty of Mathematics and Natural Sciences, South-West University "Neofit Rilski", 66 Ivan Michailov Str., 2700 Blagoevgrad, Bulgaria
[2]    Department of Aquatic Ecosystems, Institute of Biodiversity and Ecosystem Research, Bulgarian Academy of Sciences, 1 Tsar Osvoboditel Blvd., 1000 Sofia, Bulgaria
*    Correspondence: emily.varadinova@gmail.com

**Abstract:** A survey of the macrozoobenthos communities in the Maritsa River (South Bulgaria) was carried out in the summer of 2021. Benthic samples were collected and physicochemical parameters (water temperature, pH, conductivity, dissolved oxygen and nutrients) were measured at 15 sites located on the main river and its tributary system. The studied sites belonged to different river types and characterised the diversity of the ecological conditions—from unaffected to anthropogenically influenced river stretches. In addition, data from a study conducted in the summer of 2020 were used to analyse species–factor interactions in the river ecosystems and to assess the bio-indicative potential of the aquatic invertebrates. The dynamics of the taxonomic composition and abundance of the macrozoobenthos were analysed in relation to environmental factors. The physicochemical conditions of the water environment changed during the period of high water, which led to a reduction in the composition of the macrozoobenthos. Plecoptera and Trichoptera decreased in richness and abundance downstream and under human impacts. Ephemeroptera and Chironomidae were permanently present along the whole river. Oligochaeta increased in the lower river reaches and at sites with a greater amount of organic matter. The ecological status determined by the macrozoobenthos varied from high (site 1) to good, moderate and bad (site 13) at the studied sites.

**Keywords:** river; macrozoobenthos; physicochemical parameters; ecological status

## 1. Introduction

The Water Framework Directive [1] defines water not simply as a commercial resource, but rather as a heritage to be protected. Thus, the European Union water legislation has introduced a new approach for assessing the status of surface waters in order to ensure their quality and the integrity of aquatic ecosystems. In this regard, the indicative potential of key aquatic communities, including the macrozoobenthos, occupies an important place and plays a leading role in rivers.

Maritsa is the longest river that rises in the territory of Bulgaria. It has the largest catchment basin and river outflow on the Bulgarian–Greek border. The Maritsa River has been the subject of hydrobiological research since the 1960s. During this period, various studies on the taxonomic composition and biodiversity of the invertebrate communities were conducted.

The ecological situation (saprobic conditions) of the Maritsa River main stream and its tributaries have been characterised through the macrozoobenthos in a series of scientific studies [2–10].

The water quality based on physicochemical [11] and microbiological parameters [12], and organic pollutants in sediments [13], as well as the bio-indicative potential of the helminthes in the Maritsa River basin [14], have been analysed. Complex hydrobiological

monitoring of the Maritsa River catchment area based on physicochemical and benthological parameters (diversity indices, biotic index, EPT index and trophic index) was conducted by Vidinova et al. [15].

Park et al. [16] summarised thematic publications devoted to the species diversity of the macrozoobenthos in Maritsa River and some of its tributaries, presented an up-to-date list of the taxonomic composition and analysed the cenotic structure of the invertebrates.

The purpose of the present article is to analyse the macroinvertebrate communities in the Maritsa River and some of its tributaries in relation to various environmental factors and to determine the current ecological status of the studied aquatic ecosystems through the macrozoobenthic communities.

## 2. Materials and Methods

### 2.1. Study Area and Sampling Approach

The study was conducted in July 2021. The physicochemical parameters were measured and benthological samples were collected from a total of 15 sites in Maritsa River and some of its tributaries. Most of the sites (11) were located along the main river (1, 2, 4, 5, 6, 7, 8, 9, 12, 14 and 15) and the others (4) on some of the main tributaries (sites 3, 10, 11 and 13) (Figure 1). The sites belonged to four river types according to the Bulgarian river typology—R3, mountainous rivers in the ecoregion 7; R5, semi-mountainous rivers with gravel substrata in the ecoregion 7; R12, large lowland rivers with fine substrata in the ecoregion 7 and R13, small- and medium-sized lowland rivers with fine substrata in the Aegean Region [17].

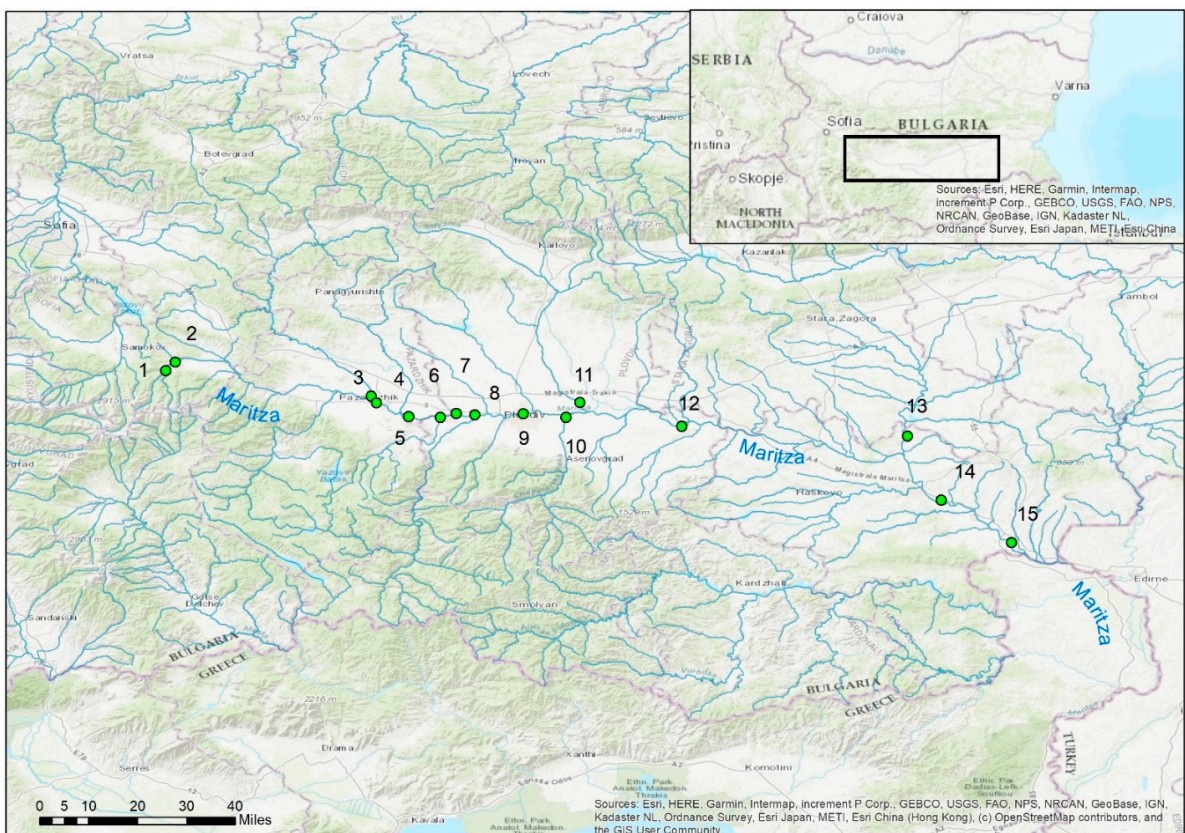

**Figure 1.** Location of the studied sites along the Maritsa River.

The characteristics (geographical coordinates, altitude, river type and sampling date) of the studied sites are presented in Table 1. A more detailed description of the studied sites, including substrate composition and land use/disturbances, was presented in [16]. The physicochemical parameters (water temperature WT, °C; pH; conductivity COND, µs/cm;

dissolved oxygen DO, mg/L; nitrates $NO_3^-N$, mg/L; and phosphates $PO_4^-P$, mg/L) were measured at the sites using a portable Windaus Labortechnik Package.

**Table 1.** Names and characteristics of the sampling sites.

| Site N | Name of the Site | Abbreviations | Geographical Coordinates (N, E) | Altitude (m) | River Type | Sampling Period | |
|---|---|---|---|---|---|---|---|
| | | | | | | 2020 | 2021 |
| 1 | Maritsa River, near the village of Raduil | M_RADUIL | 42.28165, 23.68543 | 869 | R3 | 11.08 | 23.07 |
| 2 | Maritsa River before the town of Dolna Banya | M_DBANYA | 42.30681, 23.71452 | 762 | R3 | 11.08 | 23.07 |
| 3 | Topolnitsa River, Pazardzhik, the bridge for Boshulia village, before the estuary | TOPOLNITSA_R | 42.20655, 24.29543 | 206 | R5 | 19.08 | 23.07 |
| 4 | Maritsa River, Pazardzhik, before the first bridge of the town | M_PAZARDZHIK | 42.18563, 24.31067 | 205 | R12 | 18.08 | 23.07 |
| 5 | Maritsa River, Ognyanovo Village, after Luda Yana River | M_OGNYANOVO | 42.14482, 24.40597 | 193 | R12 | 18.08 | 23.07 |
| 6 | Maritsa River, Govedare Village | M_GOVEDARE | 42.14281, 24.499538 | 195 | R12 | 17.08 | 23.07 |
| 7 | Maritsa River, Stamboliyski, the bridge | M_STAMBOLIYSKI | 42.15476, 24.54677 | 184 | R12 | 17.08 | 23.07 |
| 8 | Maritsa River, before Vacha River, landfill of Plovdiv | M_VACHA | 42.15078, 24.60184 | 171 | R12 | 17.08 | 23.07 |
| 9 | Maritsa River, Plovdiv, walkways, HMS 304 | M_PLOVDIV | 42.153513, 24.745623 | 163 | R12 | 16.08 | 24.07 |
| 10 | Chepelarska River, the bridge of Kemera | CHEPELARSKA_R | 42.14373, 24.87182 | 145 | R5 | 21.08 | 24.07 |
| 11 | Stryama River, Manole Village, bridge | STRYAMA_R | 42.18691, 24.91335 | 157 | R13 | 14.08 | 24.07 |
| 12 | Maritsa River, bridge for Parvomay, after the bridge of Parvomay, left shore before the Mechka River | M_PARVOMAY | 42.11675, 25.21602 | 140 | R12 | 21.08 | 24.07 |
| 13 | Sazliyka River, before the estuary; bridge for Svirkovo Village and Troyan Village | SAZLIYKA_R | 42.0871, 25.88515 | 79 | R13 | 16.09 | 24.07 |
| 14 | Maritsa River, after Harmanli, Complex "Gergana", HMS | M_HARMANLI | 41.89772, 25.98478 | 61 | R12 | 16.09 | 24.07 |
| 15 | Maritsa River, Svilengrad, before the old bridge | M_SVILENGRAD | 41.77254, 26.19356 | 54 | R12 | 16.09 | 24.07 |

The macrozoobenthos was collected following the multi-habitat sampling approach [18] according to the standards BDS EN ISO 10870:2012 and EN 16150:2012. The benthic samples were washed, cleaned and sorted in a hydrobiological laboratory.

Determination of the taxa was performed using identification keys [16] to a level that allowed the calculation of the indices for the assessment of the ecological status of the studied sites.

*2.2. Data Analysis and Statistical Methods*

Data regarding the taxonomic composition of the macrozoobenthos received from the survey conducted in the summer of 2020 [16] were used to analyse the species–factor

interactions and for the assessment of the ecological status of the sites during two years (2020 and 2021). The ecological status assessment was conducted based on the Biotic Index (BI) [17], which is regulated in the national water legislation requirements (Ordinance H-4/2012).

Microsoft Excel datasheets (Windows 10) were used for the visualisation of the physicochemical parameters and taxonomic composition and abundance of the macrozoobenthos at each site. The Primer 6 programme (cluster analyses, Bray–Curtis similarity and complete linkage) [19] was applied to reveal the similarity of the benthic taxa between the different sites. In the Canoco 5 package [20], canonical correspondence analysis (CCA) was used to present the species–environmental factor interactions at each site.

## 3. Results

### 3.1. Physicochemical Parameters

The dynamics of the values of the physicochemical parameters in the studied sites during the two years showed that, in 2021, higher WT, a slight decrease in pH (with the exception of site 15), lower COND and lower DO concentrations were registered compared with 2020 (Figure 2).

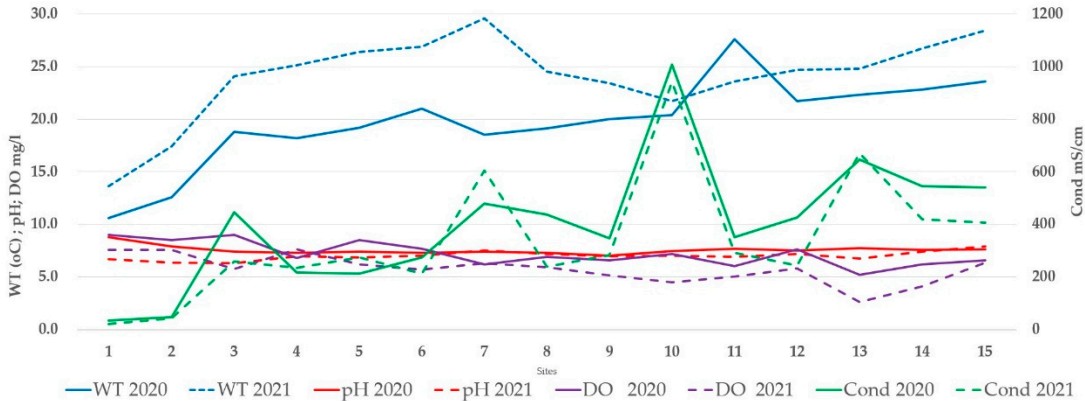

**Figure 2.** Dynamics of the WT, pH, COND and DO values at the studied sites in 2020 and 2021.

Phosphates and especially nitrates showed considerably higher concentrations in 2021 compared with 2020 (Figure 3). The values of nutrients remained almost unchanged only at the two referent sites (1 and 2).

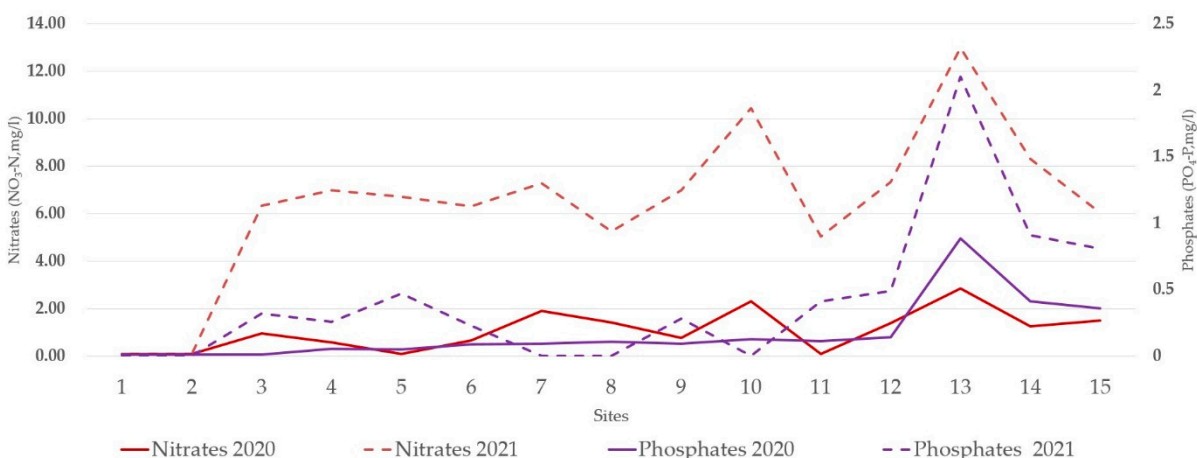

**Figure 3.** Dynamics of the values of the nutrients at the studied sites in 2020 and 2021.

### 3.2. Macrozoobenthos Community

A total of 110 benthic taxa were identified in 2021, while the number of taxa found in 2020 was 165 [16]. The greatest taxonomic richness was found at 11_2020 (35 taxa), 3_2020 (34 taxa), 12_2020 (30 taxa), 1_2020 and 2_2020 (29 taxa). A decrease in taxonomic richness was recorded even at the two unaffected mountain sites—1 (from 29 taxa in 2020 to 22 taxa in 2021) and 2 (from 29 taxa in 2020 to 17 taxa in 2021) (Figure 4). The greatest reduction in the taxonomic composition of the macrozoobenthos was registered at site 6 (from 23 in 2020 to 7 taxa in 2021). An exception was observed at site 13, where the lowest number of taxa was recorded in both years. Only two oligochaete taxa were found in 2020 and six taxa (four oligochaete worms, one mussel and one gammarid) were recorded in 2021. This site was found to have the second-highest abundance, with 1100 individuals registered in 2020. The highest abundance (1131 individuals) was recorded at site 3 in 2020 and the lowest abundance was found at 9_2021 and 10_2021 (only 34 specimens). The total abundance decreased by almost four times in 2021 (2363 specimens) compared with 2020 (8461 specimens) (Figure 4).

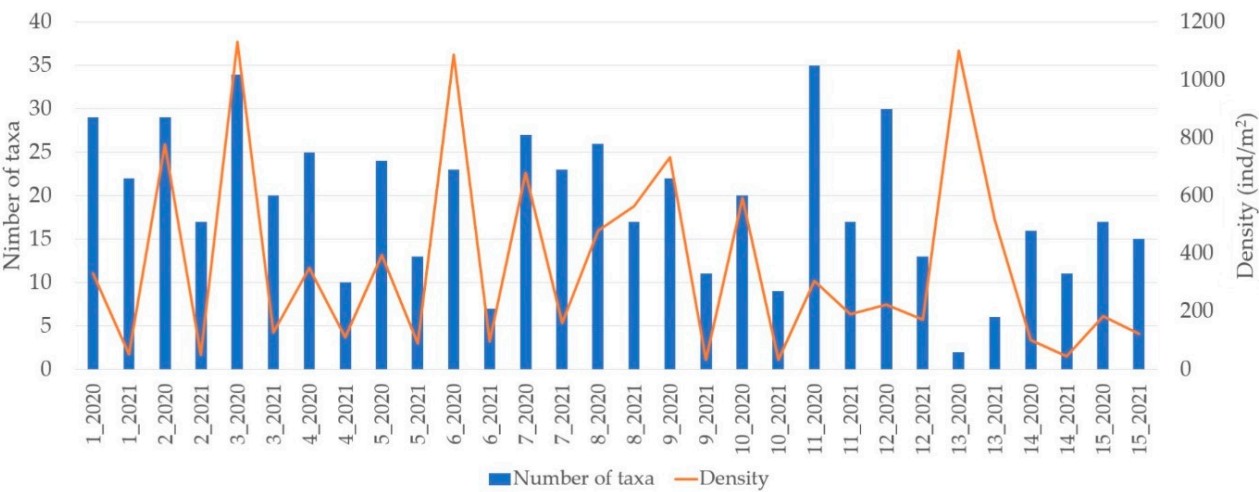

**Figure 4.** Number of macroinvertebrate taxa and density (ind./m$^2$) at the studied sites.

The distribution by main taxonomic groups showed that the order Ephemeroptera was present at almost all studied sites. Representatives of this group were not found at the heavily impacted site 13 during both study periods, at site 15 in 2020 and at site 10 in 2021 (Figure 5).

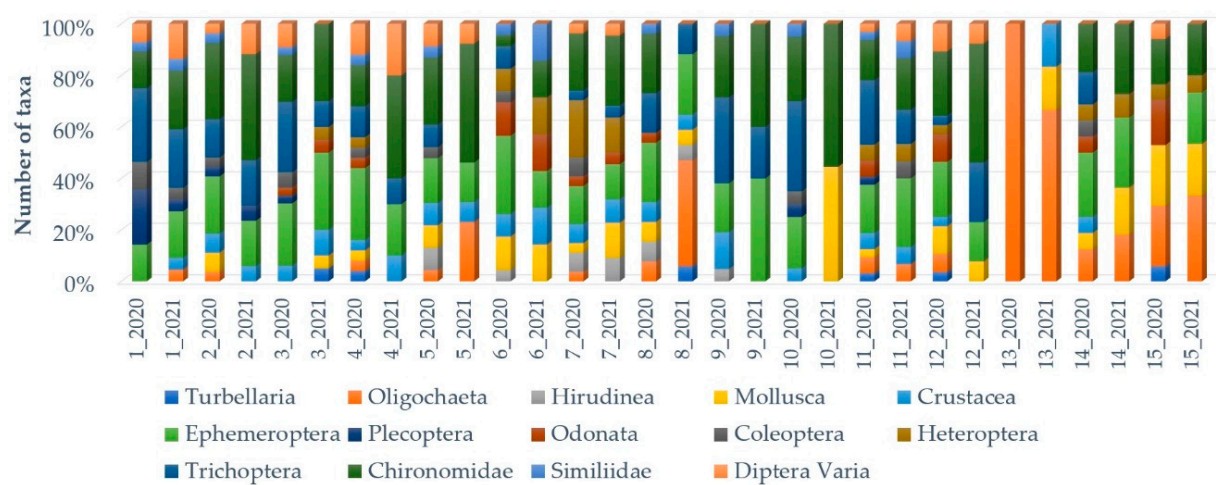

**Figure 5.** Percentage ratio of the major taxonomic groups at the studied sites in 2020 and 2021.

The orders Plecoptera and Trichoptera prevailed at the cleanest site, site 1. The numbers of stoneflies decreased downstream to complete absence at the lower river stretches and at the polluted sites. Caddisflies had greater taxonomic richness in 2020. They were not found in the composition of the benthic communities at site 14 in 2021, or at sites 13 and 15 in both years. Oligochaeta worms, which are more adaptable to changes in the aquatic environment, predominated in the lower river reaches at the anthropogenically influenced sites. The family Chironomidae was persistently present throughout the river during both study periods, except at site 13 and site 8 in 2021.

A considerable reduction in Ephemeroptera, Plecoptera and Trichoptera densities were recorded in 2021 compared with 2020 (Figure 6). Ephemeroptera abundance showed a slight decreasing trend downstream. In 2020, the density of Plecoptera was the largest at the first three (mountainous and semi-mountainous) sites of the main stream (1, 2 and 3), and at sites 10 and 11, which were situated in the tributary system. Stoneflies were poorly represented or absent downstream. The abundance of oligochaetes increased in the lower river reaches, with the highest numbers found at site 13 (1100 specimens). The abundance of Chironomidae was relatively permanent at the studied sites (Figure 6).

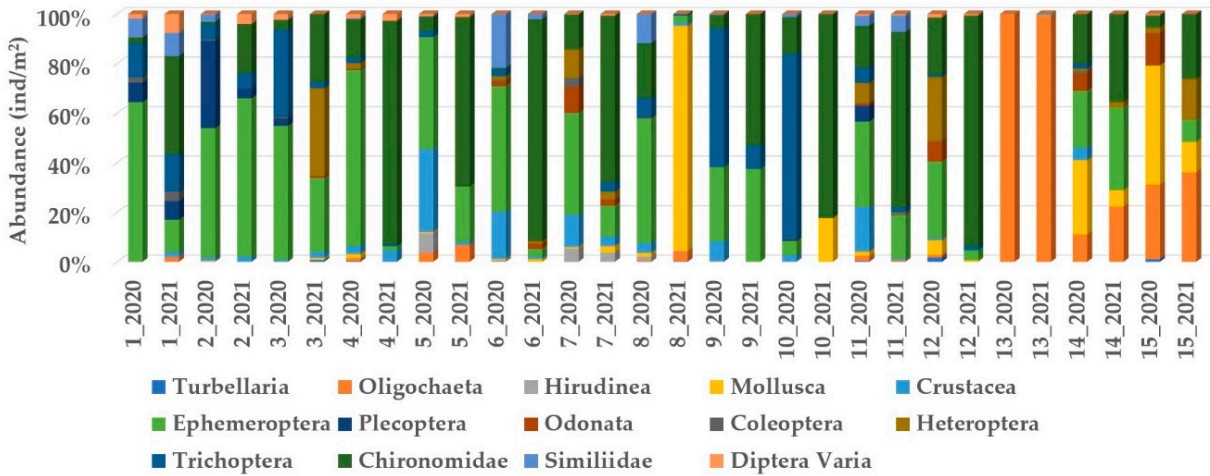

**Figure 6.** Percentage ratio of the abundance of the major taxonomic groups at the studied sites in 2020 and 2021.

Cluster analysis revealed more similarity between samples collected from different sites in the same year. Thus, greater resemblance in taxonomic composition was observed at sites 3, 4, 5, 6, 7, 8, 9 and 10 in 2020 than between the samples from the same sites collected in the two years. The taxonomic composition at site 13 was very poor, composed mainly of oligochaete worms in both years. The unpolluted sites (1 and 2) formed one cluster, but again the leading factor that determined the similarity between the samples was the year of study (Figure 7).

The relationship between the macrozoobenthos and measured environmental factors was analysed using CCA (F = 3.7791, $p$ = 0.002) (Figure 8). Axes 1 and 2 are presented, as they cumulatively account for 65, 53 % of the total variance. The species-environmental correlations of each axis were 0.95 (axis 1) and 0.84 (axis 2) (Table 2). Among the physicochemical variables, DO (−0.679) and $PO_4^-P$ (0.859) exhibited the strongest correlation with the first axis. WT (0.646) and $NO_3^-N$ (0.523) had the best correlation with the second axis. Ologochaeta (2.116), Diptera (0.567) and Plecoptera (0.517) had the highest correlation with the first axis, as well as Mollusca (1.853). Plecoptera (−1.153) and Trichoptera (0.706) exhibited the strongest correlation with the second axis. There was a clear separation of the samples between the two years of study (Figure 8a). The only exceptions were the two unaffected sites, 1 and 2, located at the upper parts of Maritsa River, where no considerable differences in environmental conditions were observed between the two studied periods. Taxonomic groups that included species with preferences for higher oxygen concentrations

and that are more sensitive to changes in the aquatic environment were located in the lowest left part of the ordination diagram (Figure 8b). The groups that contained taxa more tolerant to nutrient loading were situated in the upper right part of the ordination space, which displayed the gradient of increasing nitrate and phosphate concentrations.

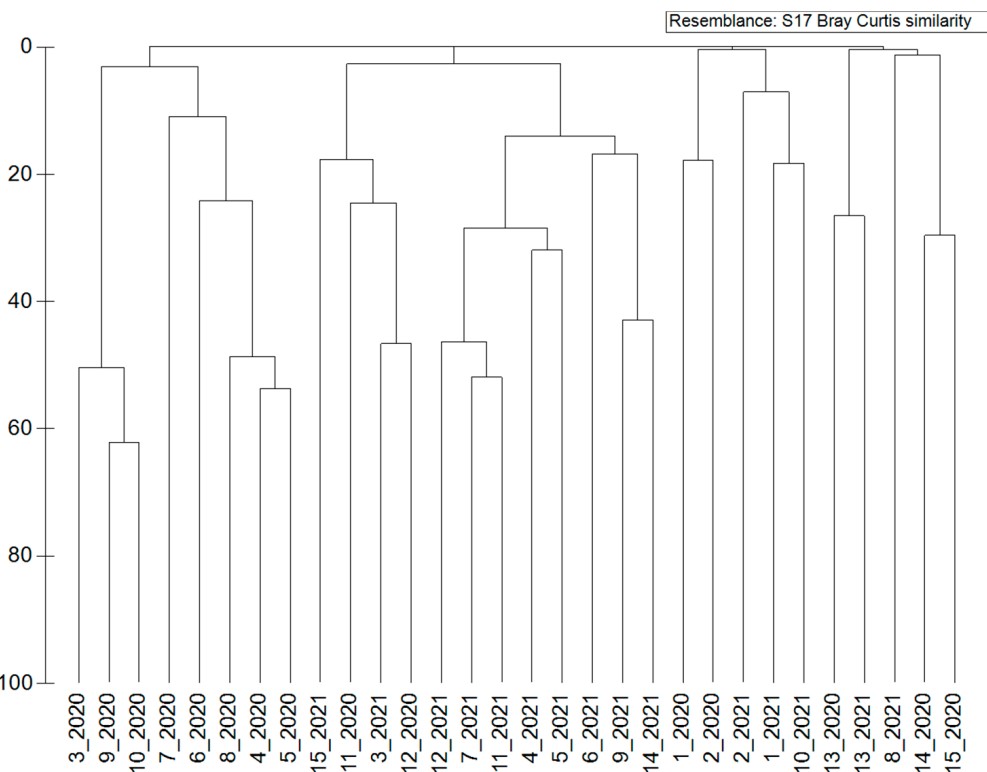

**Figure 7.** Cluster analysis of the similarity in the taxonomic composition between the studied sites in 2020 and 2021.

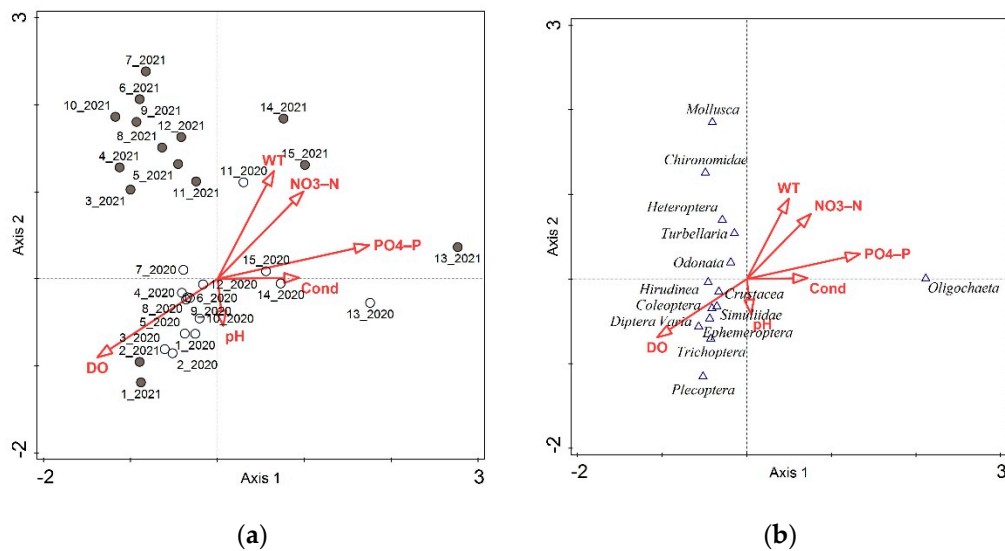

**Figure 8.** CCA of the physicochemical factors of the studied sites (**a**) and taxa (**b**). (The red arrows show the distribution of environmental factors in the ordination space)

**Table 2.** Eigenvalues, variations and correlation of the CCA.

|  | **Axis 1** | **Axis 2** | **Axis 3** | **Axis 4** |
|---|---|---|---|---|
| Eigenvalues | 0.8105 | 0.4016 | 0.2217 | 0.0767 |
| Explained variation (cumulative) | 26.27 | 39.29 | 46.47 | 48.96 |
| Pseudo-canonical correlation | 0.9483 | 0.8423 | 0.7906 | 0.5557 |
| Explained fitted variation (cumulative) | 52.92 | 79.13 | 93.61 | 98.62 |

*3.3. Ecological Status Evaluation*

The assessment based on the macrozoobenthos showed that the ecological status was unchanged at sites 1, 2, 3, 7, 8, 11 and 13 during the studied period (Figure 9). The value of BI decreased by half a unit at sites 2 and 3, but the ecological status remained within the limits of the good status. Although slightly improved, the ecological status at site 13 remained bad and deteriorated by one grade—from good to moderate—at sites 4, 5, 6, 9, 10, 12, 14 and 15.

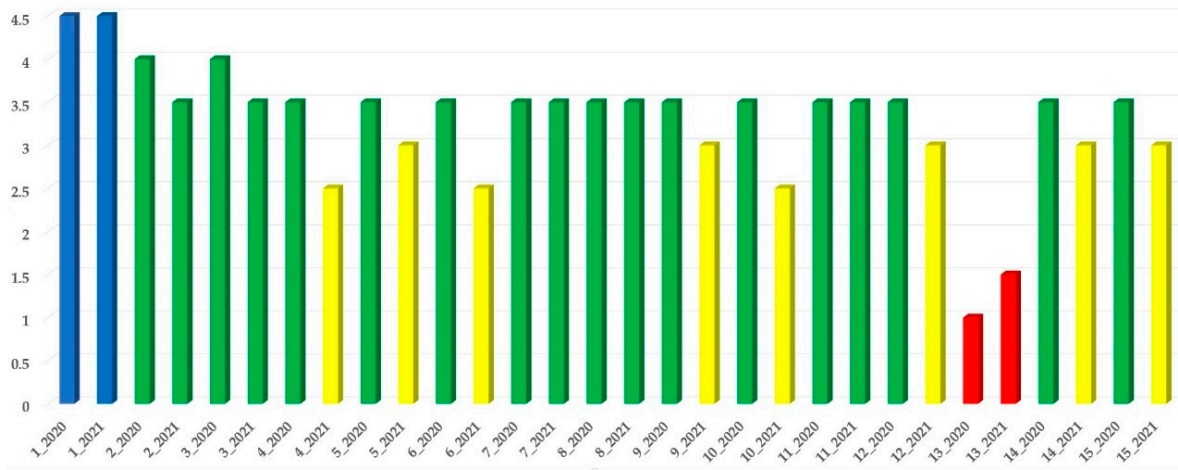

**Figure 9.** Ecological status assessment of the studied sites (blue, high; green, good; yellow, moderate; red, bad).

## 4. Discussion

The variability in the biotic data at different sites was due to various factors, among which substrate type, velocity, depth and altitude had a structure-defining role. According to Thoker et al. [21], high diversity in natural pristine rivers at a high altitude is related to low stress, while the low diversity at lower courses signifies environmental stress due to human activities. The studied sites in the Maritsa River belonged to different river types, some characterised as natural and others as highly modified water bodies [16]. Various conditions of the water environment were observed along the river course as a result of external factors—physical and anthropogenic. Our results demonstrated a considerable decrease in the taxonomic richness at all studied sites in 2021 in comparison with 2020 (Figure 4.) More taxa were dropped from the benthic community in the lower reaches of Maritsa River. In 2020, Park et al. [16] recorded the highest taxonomic richness at the upper course of the river (mountainous sites 1 and 2—29 taxa), at the slightly affected section (site 12—30 taxa) and at the tributary Stryama River (site 11—35 taxa). In comparison, in 2021, the taxonomic richness decreased, with 7 taxa at site 1, with 12 taxa at site 2, with 17 taxa at site 12 and with 18 taxa at site 11. The greatest reduction in the taxonomic composition was found in the orders Ephemeroptera, Trichoptera and Plecoptera (Figure 5). Furthermore, a decrease in the number of taxa was registered even at the reference sites (1 and 2), which were characterised as undisturbed or with very minor human impact.

Krepski et al. [22] reported that the hydrological factors have a significant impact on the development of aquatic biocenoses, and the greatest biodiversity and the highest abundance of zoobenthic organisms were noted under the lowest water flow rates. In 2021, the survey was conducted at the end of July, when higher waters in Maritsa River were observed in comparison with 2020. In 2021, the water environmental conditions were characterised with higher WT, lower pH, greater COND and lower DO concentrations at the studied sites (Figure 2). The complex action of these environmental factors formed radically different conditions during the two study periods, which structured different benthic communities. This was evident from the clear separation of the sites studied in 2020 and 2021 (Figures 7 and 8). It should be noted that high waters shifted the littoral zone inland and enhanced the biological drift. The displacement of macroinvertebrates through the water column is a natural process that can lead to massive translocation downstream, especially under high discharge [23]. According to Kurbanov et al. [24], a well-known depression of the zoobenthos communities is observed during the flood period. Therefore, the higher waters registered during the field observation in 2021 probably created a more unstable aquatic environment and caused habitat alteration and/or degradation, which was a prerequisite for the decrease in the taxonomic richness at the studied sites. Under such less-favourable conditions, the more tolerant family Chironomidae was permanently present in Maritsa River and the studied tributaries with greater abundance in the middle and lower sections of the stream. As Čerba et al. [25] reported, the taxa of this group differed depending on the substrate type and showed preference and adaptation to microhabitats with specific conditions and food availability.

According to Brysiewicz et al. [26], the oxygen and nitrate-nitrogen contents and water temperature have the greatest effects on the various groups of macrozoobenthos. Our results revealed high negative correlations of the DO with WT ($-0.640$), $NO_3^-N$ ($-0.775$) and $PO_4^-P$ ($-0.826$). The concentrations of nutrients (nitrates and phosphates) showed increased values in the high-water period (2021). $NO_3^-N$ repeatedly exceeded the values measured in 2020 at all studied sites, except at reference sites 1 and 2. The presence of large amounts of nutrients could be a result of the accumulation of allochthonous inputs, collected longitudinally from the catchment area and from the larger floodplain coastal areas formed during the periods of high water. The highest values were observed at anthropogenically influenced site 10 and the most affected site 13, where Park et al. [16] registered considerable agricultural activities (Figure 3). Anthropogenic alterations affect habitat quality, decrease species diversity and increase the dominance of pollution-tolerant taxa [21]. At site 13, a very small number of pollution-resistant taxa (mainly oligochaetes) (2020—two taxa; 2021—six taxa) represented by a large number of individuals were recorded in both years. Both samples (13_2020 and 13_2021) were located in the gradient formed by the COND and nutrient vectors on the ordination diagram (Figure 8a).

The composition of macrozoobenthos is mediated by substrate availability, water chemistry and the availability of nutritional resources [27]. The formation of the specific conditions registered in 2021 had led to a transformation of the aquatic invertebrate communities, in which the more sensitive benthic species dropped out of the macrozoobenthos under increased environmental stress. Such a transformation was noticed to some extent in order Ephemeroptera, the abundance of which considerably decreased downstream. Stoneflies and trichopterans decreased in species richness and abundance downstream and under human impact in both years, and the reduction was more pronounced in 2021 (Figures 5 and 6). Ephemeroptera, Plecoptera and Trichoptera taxa are potentially sensitive to changes (especially increasing disturbances) and cannot tolerate any presence of pollutants in the water bodies [28]. This was also confirmed by the CCA diagram, on which these groups were located in the ordination space around the DO gradient. Anthropogenic pollution causes changes in the composition and structure of aquatic communities, expressed in the change in the dominant complexes of organisms, the simplification of the ecological structure and the appearance of highly saprobic species in the dominant complexes [24]. Therefore, as they are more tolerant towards pollution and are an indicator of changes in

aquatic habitats, oligochaete worms increased in terms of taxa richness and abundance at the organic-laden sites.

Since the studied sites were of four different types (R3, R5, R12 and R13) according to the national typology system [29], it was possible to analyse various ecological situations and to conduct a complex and relevant ecological assessment of the studied river system through the macrozoobenthos. Owing to their indicative capacity, macroinvertebrates have been defined as an obligatory biological quality element according to the requirements of the European Union and national water legislation. The species diversity and community pattern of macrozoobenthic invertebrates are used to evaluate the environmental stress resulting from a variety of anthropogenic disturbances [21]. The aquatic conditions changed under human impact were reflected in the composition and structure of the macrozoobenthos, as indicated by the BI values and the corresponding evaluation of the ecological status. Thus, the lowest BI values (BI 2020—1; BI 2021—1.5), which defined poor ecological status, were observed at site 13 (Figure 9). The values of BI only remained unchanged for sites 1, 7, 8 and 11 (BI 3.5), and although a reduction in the number of taxa was recorded in 2021, these sites maintained good ecological status during both years. Decreased values of BI were registered at unaffected site 2 and at slightly polluted site 3, but the assessment indicated the same good ecological class (Figure 9).

A deterioration in the ecological situation at the rest of the studied sites was recorded in 2021. It should be noted that considerable anthropogenic impact was registered at these sites in 2020, as a result of hydromorphological pressure, agricultural impact and the urbanisation of the territories around Maritsa River [16]. These negative effects on the water ecosystems were confirmed during the sampling in 2021. Kirin [13] reported that the sediments were loaded with organic pollutants at the Maritsa River sections in which sites 9, 10, 11 and 12 were located. The higher water quantities recorded in 2021 had probably caused sediments to rise from the substrate and to further contaminate the river. Thus, pollution had worsened the environment where aquatic invertebrates live, causing the replacement of some sensitive community taxa with pollution-tolerant ones. These processes might also contribute to the ecological status decline of the river ecosystems at these sections in 2021 (Figure 9). The registered moderate ecological status of the studied river sections requires scientifically based decisions and the undertaking of adequate measures to improve the conditions and achieve good ecological status.

## 5. Conclusions

The structuring of the composition of the macroinvertebrate communities in Maritsa River occurred under the influence of natural and anthropogenic factors. The hydrological regime with the complex action of the physicochemical parameters of the water environment were of vital importance for the diversity and abundance of the macrozoobenthos. The human impact altered the habitat characteristics and ecosystem integrity, and reflected on the ecological situation in the studied river sections. This led to a transformation in the taxonomic groups and in the benthic community as a whole. At the affected sites and with the accumulation of nutrients downstream, the invertebrate communities were represented by more tolerant groups. Thus, macrozoobenthic communities had good bio-indicative potential for the assessment of the ecological status of the lotic ecosystems.

**Author Contributions:** Conceptualization, E.V. and L.S.; methodology, E.V. and L.S.; formal analysis, E.V.; investigation and sample collection, M.I. and J.P.; sample identification and sorting, E.V., J.P.; data curation, M.I., J.P. and V.T.; writing—original draft preparation, E.V.; writing—review and editing, E.V., L.S.; visualisation, E.V., M.I., J.P. and V.T.; statistical analysis, V.T.; supervision, E.V. All authors have read and agreed to the published version of the manuscript.

**Funding:** The study was funded by the World Bank through the project "Validation of the typology and classification system in Bulgaria for the ecological status assessment of the surface water bodies in categories "river", "lake" and "transitional waters"" (Grant no. 71 957 35/17.4.2020, DICON-UBA).

**Institutional Review Board Statement:** Not applicable.

**Data Availability Statement:** The data presented in this study are available in article.

**Acknowledgments:** The article is published with the financial support of the SWU "N. Rilski" Blagoevgrad.

**Conflicts of Interest:** The authors declare no conflict of interest.

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
