# Peer review of "Characterisation of Macroinvertebrate Communities in Maritsa River (South Bulgaria)—Relation to Different Environmental Factors and Ecological Status Assessment"

_diversity, doi:10.3390/d14100833_

Round 1

Reviewer 1 Report

The current manuscript reports a study conducted in the Maritsa River, located in the South of Bulgaria, where the authors made two macrozoobenthos surveys in two consecutive years, with the concomitant measurement of environmental parameters. Although the subject of the study is not new, neither the methods nor the data analyses conducted, it adds information of the quality status of 15 sites in this Bulgarian region, which can be relevant in a WFD perspective. The manuscript is objective and well written, although some aspects can be improved. The discussion is also well done and supported by the results obtained. Below the authors can find some minor suggestions per section.

Abstract

Replace “situated” by “located”

Replace “situations” by “conditions”

Keywords: I would suggest that the authors also add to the keywords “River” or “Freshwater Ecosystems”

Introduction

L51: I know that for the Diversity journal the references are cited throughout the text in the format of numbers. However, it would be easier for the readers if the authors in this cases also display the reference with the authors name, such as Vidinova et al. [15].

L52: the same comment made earlier in L51 also applies here and throughout the text (for example in the discussion)

L58: I would suggest that the authors add “communities” after “macrozoobenthos”

Materials and Methods

L77: the “3” should be under scripted and a superscripted “-“ should be added after. Please alter here and throughout the text and make the adequate alterations for the remaining nutrients.

L89: please provide the identification keys used and be more specific with the identification level used.

L98: do you rather mean Excel datasheets, graphics?

Results

Figure 2 and all figures: please increase the letter size of the numbers in the axes and legends

L122-132: this is a very generalist description of the results. I would suggest that the authors give some more few details such as for example, differences between sites such as the sites displaying the lowest diversity and abundance and sites displaying the highest diversity and abundance, etc.

L152: replace “Figute” by “Figure”

L168: replace “Fugure” by “Figure”

L180: I would suggest that the authors replace “situated” by “located”

L184-186: I would suggest that the authors rephrase this sentence to something like “The groups that contained taxa more tolerant to nutrient loading were situated in the upper and right part of the ordination space, which displayed the gradient of increasing nitrates and phosphates concentrations.”

Figure 8: Why the authors did not join all results in the same diagram: taxa, sites and environmental factors? Because by using CCA the authors can do this. If the option was to not present a crowded figure, the authors should somehow justify their choice. If  the authors opt to maintain the analyses, at least, the CCA axes should have the same number scales. It will make the comparison easier, in order to check the taxa associated with specific sites, environmental variables etc.

Discussion

L225: please replace “comparision” by “comparison”

L230: remove the “.” before “(Figure 7..”

L239: replace “grater” by “greater”

L243: nitrate and nitrogen contents?

L263: replace “extend” by “extent”

L277: add “Since” at the beginning of the sentence

Reviewer 2 Report

First, thank you for your suggestion to review this manuscript entitled "Characterization of Macroinvertebrate communities in Maritsa River (South Bulgaria). Relation to different environmental factors and ecological status assessment.

It is a very interesting manuscript, well written and presenting the research results in an interesting way.

My minor remarks:

- it is a pity that the authors did not do a longer period of research (e.g. comparison between seasons) because in this way some research becomes too regional. Has Maritsa not changed in recent years (e.g. climate change, anthropopressure)? Such a comparison would be interesting and if it is not possible now, I encourage you to undertake such research in the future.

- picture 1 is a bit too bright - the river is not visible.

- suggests in Table 1 to insert a short description of the catchment area in a given section of the river

- in figures 2-6, please increase the font, as it is a bit too small.

All in all, a very good manuscript which I recommend for publication in the journal "Diversity". Written in good language, the results are clearly presented and you can see that the authors did quite a bit of work into this manuscript.

Congratulations to the Authors and encourage them to do further research.
